# An Overview of Methods and Exemplars of the Use of Mendelian Randomisation in Nutritional Research

**DOI:** 10.3390/nu14163408

**Published:** 2022-08-19

**Authors:** Derrick A. Bennett, Huaidong Du

**Affiliations:** 1Clinical Trial Service Unit and Epidemiological Studies Unit (CTSU), Nuffield Department of Population Health, University of Oxford, Oxford OX3 7LF, UK; 2Medical Research Council Population Health Research Unit, University of Oxford, Oxford OX1 3QR, UK

**Keywords:** genetics, mendelian randomisation, causal, non-communicable disease, biomarkers, exposures

## Abstract

**Objectives:** It is crucial to elucidate the causal relevance of nutritional exposures (such as dietary patterns, food intake, macronutrients intake, circulating micronutrients), or biomarkers in non-communicable diseases (NCDs) in order to find effective strategies for NCD prevention. Classical observational studies have found evidence of associations between nutritional exposures and NCD development, but such studies are prone to confounding and other biases. This has direct relevance for translation research, as using unreliable evidence can lead to the failure of trials of nutritional interventions. Facilitated by the availability of large-scale genetic data, Mendelian randomization studies are increasingly used to ascertain the causal relevance of nutritional exposures and biomarkers for many NCDs. **Methods:** A narrative overview was conducted in order to demonstrate and describe the utility of Mendelian randomization studies, for individuals with little prior knowledge engaged in nutritional epidemiological research. **Results:** We provide an overview, rationale and basic description of the methods, as well as strengths and limitations of Mendelian randomization studies. We give selected examples from the contemporary nutritional literature where Mendelian randomization has provided useful evidence on the potential causal relevance of nutritional exposures. **Conclusions:** The selected exemplars demonstrate the importance of well-conducted Mendelian randomization studies as a robust tool to prioritize nutritional exposures for further investigation.

## 1. Background

Nutritional epidemiology primarily aims to elucidate the roles of nutritional factors (such as dietary patterns, food intake, macronutrients intake, circulating micronutrients and metabolites), in the onset of non-communicable diseases (NCDs) such as cancer, cardiovascular disease, and diabetes [1]. These NCDs are responsible for a great deal of mortality and morbidity as measured by disability adjusted life years and years of life lost [2]. Over the past two decades, large-scale observational studies have identified many potential modifiable nutritional factors of NCDs [3,4,5,6]. However, these findings have failed to be replicated in randomized trials (e.g., beta-carotene and cancer risk), and the lack of replication has been attributed to an inability to control for confounding in the observational studies [7]. A substantial proportion of nutritional findings based on observational studies has therefore been very strongly criticized as being biased due to extensive residual confounding based on poor measurements of dietary exposure, modest effect sizes and selective reporting [8].

One key issue is that nutritional and lifestyle factors often occur together or are highly correlated, (e.g., people with good diets often engage in other healthy lifestyle activities). Although several analytic strategies are available to control for confounding, unmeasured and residual confounding often remain a concern in nutritional epidemiology, in common with other areas of observational research [9]. Another key issue of concern for observational research in general is reverse causality in which preclinical or incipient disease influences the exposure status rather than vice-versa. This is particularly a problem in studies of behavioural exposures such as nutrition where an individual may have changed their behaviour due to diagnosis of a particular disease [10].

There is a huge literature on the measurement of nutritional factors, but self-reported nutritional status assessment is known to be problematic, and this means that there is a need for analytical determinants that can objectively and accurately quantify nutritional status. Nutritional biomarkers provide a more proximal measure of nutrient status than dietary intake and can be directly measured in buccal samples (e.g., blood, urine). These nutritional biomarkers can be used as indicators of nutritional status with respect to the intake or metabolism of dietary constituents. Table 1 give some examples of potential biomarkers for assessment of nutritional intake.

Identification of efficacious therapies in large-scale randomized controlled clinical trials (RCTs) have contributed to the reduction of NCDs. RCTs are the “gold-standard” for inferring the causal role of an exposure in the development of disease. However, even though RCTs can provide more rigorous evidence for causality, long-term RCTs of nutritional interventions with disease outcomes are often not feasible due to the inability to identify appropriate control groups and to blind participants and researchers, and poor compliance [11]. Due to these difficulties, alternative non-interventional or quasi-randomized approaches have been increasingly used in recent times in clinical research [12,13], particularly in areas where a strong RCT evidence-base is absent. One such analytical approach with has gained traction is one where a genetic variant, (either single or a combination of multiple genetic variants), is used to conduct “Mendelian randomization (MR)”, and a simple search of PubMed demonstrates that these studies have become increasingly common, in nutritional related research recent decades (Figure 1).

This has enabled more robust genotyping platforms analyzing millions of single nucleotide polymorphisms (SNPs), and the availability of big data from global genetic consortia, which collectively have contributed to better understanding of the genetic architecture of NCDs [14,15,16,17]. These collaborative efforts have led to the discovery of SNPs that associate with dietary patterns, macronutrient intake and plasma biomarkers (such as caffeine or copper levels), and that can then be used in MR analyses to test the causal relevance of these biomarkers in disease risk.

The primary aim of this overview is to act as a primer for nutritional researchers interested in conducting their own MR analyses. The overview also aims to explain the rationale for MR, describe the advantages and potential limitations of this type of study design, and provide some selected examples of how MR has been used in, and benefitted, nutritional research.

## 2. Principles of a Mendelian Randomization (MR) Study

In lay terms, a MR study is one in which genetic variants are used as instrumental variables (IV) [18] in order to investigate the causal relationship of a putative risk factor or biomarker and risk of disease [19,20]. The fundamental principle of MR is that, if genetic variants that either alter the level of, or imitate the biological effects of, a modifiable nutritional exposure (such as dietary pattern, macronutrient intake, or circulating micronutrients), or a biomarker that is causally related to a disease, then it is expected that these genetic variants should also be associated with disease risk to the extent predicted by the effect of the genetic variant on the level of nutritional exposure or the biomarker of interest [20].

The name “Mendelian randomization” refers to the random assortment of alleles where DNA is transferred from parent to offspring, a process named Mendel’s second law [21]. The inheritance of any particular genetic variant in an individual’s DNA should be independent of other characteristics. An example of this is selenium status, which is associated with an increased risk for various chronic diseases when levels are low [22]. It is known that the biological effects of selenium are largely mediated by a family of around 25 proteins, which contain at least one selenium-containing amino acid, selenocysteine [23]. Evidence suggests that individual requirements for selenium differ due to genetic variants in seleno-protein encoding genes. An example of how a MR study design can be considered as analogous to a parallel RCT design using selenium as an example is shown in Figure 2 and some authors have referred to MR studies as “nature’s randomized trials”.

There are three key assumptions for the reliable conduct of valid MR study (Figure 3) and these are: (i) the genetic variant(s) associates with the level of exposure of interest (the relevance assumption); (ii) the genetic variant(s) is not associated with confounders of the exposure and outcome (the independence assumption); (iii) the genetic variant(s) only influences risk of disease through the exposure of interest and not through any other pathways (the exclusion restriction assumption).

Severe departure from any of these assumptions can lead to a biased estimate, leading to an unreliable causal estimate.

The key strengths of a MR study are that SNPs are: (a) non-modifiable, and therefore less susceptible to potential “reverse causality”; (b) are unlikely to be influenced by confounding, due to Mendel’s second law; (c) are less prone to random or systematic measurement error. Essentially, MR can mitigate the main sources of bias encountered in classical observational nutritional epidemiology and provide estimates of the potential causal relevance of an exposure or biomarkers with risk of disease. MR can be applied using individual participant data (known as one-sample MR) or with summary-level data (known as two-sample MR).

## 3. Threats to the Reliability of MR

Although MR studies are better at uncovering causal relationships than classical observational epidemiological studies, they are not a panacea. There are several issues which can limit the ability to reliably determine causality which are now briefly described, and a fuller description is available elsewhere [24].

### 3.1. Linkage Disequilibrium

It is known that genetic variants located close to each other on a chromosome may be inherited together. This leads to correlation between genetic variants because the allele of one locus is disproportionately co-inherited with an allele at another locus (this is called linkage disequilibrium (LD)). Genetic variants in LD do not satisfy Mendel’s second law of random assortment, therefore use of several genetic variants in LD with each other may introduce bias in a MR study. In addition, if several genetic variants, which are in LD and are associated with the same risk factor, are used, they may overestimate the magnitude of the causal estimate. This can be avoided by the use of clumping, whereby only a single SNP is retained per locus based on LD within that population (or by using a representative reference panel); or the estimation procedure is performed after conditioning on the LD structure between genetic variants [25].

### 3.2. Population Stratification

If the population under investigation is not homogenous, but rather based on two or more substrata (e.g., different ethnic ancestries), any disease that has a higher prevalence in one of these subpopulations may incur relationships with SNPs that are predominantly found in this group. Many genome-wide association studies (GWAS) and MR studies try to minimize the impact of population stratification by including participants of just one ethnicity and by adjusting for genetic principal components that aim to capture different sub-groups within that population sample [25].

### 3.3. Inadequate Statistical Power

Most genetic variants only have a modest effect on a given exposure or biomarker of interest (i.e., they only explain a small amount of the variance). This lack of variance explained lowers statistical power and therefore very large numbers of cases are required. It is well known that statistical power can be increased by combining several SNPs into a genetic risk score (GRS), which usually increases the proportion of variance explained in the exposure or biomarker of interest. Furthermore, weighting the SNPs by their associations with the nutritional exposure or biomarker of interest from published GWAS provides additional statistical power [26].

### 3.4. Weak Instrument Bias

Good genetic instruments are SNPs that are reliably associated with the exposure from well-conducted GWAS, typically involving detection in a discovery sample at a GWAS threshold of statistical significance (e.g., *p* < 5 × 10^−8^) followed by replication in an independent sample. It is important to ensure that instruments selected for an exposure are independent of each other, and there is published guidance on good instrument selection [27]. If the association between a genetic variant and the exposure of interest is not sufficiently strong, so-called “weak instrument bias” can arise. The F-statistic [25,28], can be used to quantify the strength of the relationship between the SNPs and the outcome of interest.

### 3.5. Associations of the Genetic Variants with Other Traits: Confounding and Pleiotropy

The genetic variant or GRS should only be associated with the exposure (and its pathway) under investigation, otherwise it may not be valid to use this genetic variant or a GRS of multiple genetic variants in a MR study [24]. A genetic variant or GRS may associate with other potential exposures or confounders, a phenomenon known as “pleiotropy”. When those potential confounding traits are on other discrete pathways to the exposure of interest, this is termed horizontal pleiotropy [27] and use of the genetic variant in this circumstance may result in a unreliable estimate from the MR study (Figure 4a). Another form of pleiotropy is “vertical pleiotropy” which occurs when the genetic variant associates with other traits downstream of the main exposure of interest. For example, a variant that is associated with total cholesterol may also be associated with cholesterol fractions such as low-density lipoprotein cholesterol or high-density lipoprotein cholesterol, thus exhibiting vertical pleiotropy (Figure 4b). MR analyses are robust to this type of pleiotropy [29]. A more comprehensive description of how horizontal and vertical pleiotropy can occur and the potential impact on the interpretation of the analyses is provided elsewhere [24].

## 4. Estimating the Causal Effect in MR

In one-sample MR, if the aim is to estimate the causal effect of an exposure X on a continuous outcome Y using (one or more) genetic instruments Z for X, this is usually achieved via a two-stage least squares analysis whereby, in the first step, the exposure X* which is independent of the confounders is estimated via the genotypes of the instruments by calculating the fitted values from the regression of X on Z. In the second step, the causal effect estimate is obtained by regressing Y on X*. As both steps are performed in a single model instead of two separate regressions, the variation of both Z and X* is appropriately accounted for and thus produces the correct standard errors [18,30]. When the outcome of interest in a one sample MR is dichotomous, the exposure X is regressed on the instruments Z and the residuals of the regression are saved in the first step. In the second step, a logistic regression of the outcome Y on X is performed, adding the residuals from the first step as a covariate to the second stage model [31].

In two-sample MR, the SNP-exposure effects and the SNP-outcome effects are obtained from separate studies. With these summary data it is possible to estimate the causal influence of an exposure on the outcome. As a consequence, causal inferences can be made between exposure and outcome even though they have not been measured in the same set of samples, thus enabling the leverage of large-scale GWAS data to enhance statistical power.

The simplest way to generate a MR estimate for a single genetic variant is to divide the SNP-outcome estimate (GY) by the SNP-exposure estimate (GX) to derive the Wald ratio [26]. To obtain an MR estimate using multiple SNPs, the simplest approach is to perform an inverse variance weighted (IVW) meta-analysis of each Wald ratio (i.e., GY/GX). Alternatively, a weighted regression of the SNP-exposure estimates against the SNP-outcome estimates, with the regression constrained to pass through the origin and the weights derived from the inverse of the variance of the SNP-outcome effects [26]. When multiple SNPs are used it is assumed that the SNP-exposure and SNP-outcome association estimates are uncorrelated (so that covariance terms can be ignored) and the SNP-exposure association is measured precisely (the NO Measurement Error in the exposure (NOME) assumption) [32].

Recent advances in MR methodology have led to several sensitivity analyses being proposed for two sample MR that include MR-Egger, which can quantify the amount of bias due to horizontal pleiotropy and can provide a valid causal estimate even in the presence of horizontal pleiotropy [33]. The MR-Egger modifies the IVW analysis by allowing a non-zero intercept, thereby allowing the net-horizontal pleiotropic effect across all SNPs to be unbalanced, or directional. The method returns an unbiased causal effect even if the “no horizontal pleiotropy” assumption is violated for all SNPs but assumes that the horizontal pleiotropic effects are not correlated with the SNP-exposure effects (this is known as the Instrument Strength Independent of Direct Effect (InSIDE) assumption) [33]. Other sensitivity analyses for two sample MR include the weighted median assumes that only half of the SNPs are valid instruments (i.e., satisfy the three MR assumptions) [34]; weighted mode (clusters SNPs into groups of similarity of causal effects and produces a valid estimate if the cluster with the largest number of SNPs are valid instruments) [35]; and MR-PRESSO (identifies outlier SNPs that may have horizontal pleiotropic effects) [36]. A more detailed overview of common methods [37] and different types of sensitivity analyses for assessing pleiotropy in MR studies are discussed elsewhere [32]. A recent innovation has proposed the concept of Robustness Values (RV) that can be utilized to reveal the minimal strength of violations necessary to explain away the MR results [38]. This approach has some similarities to the E-value introduced for assessing residual confounding in conventional epidemiology [39]. Table 2 briefly summarizes some selected useful statistical software resources that are available for MR.

## 5. Extensions to Standard MR Approaches

There are also more complex extensions to standard MR, such as factorial MR (that allows the exploration of separate and joint causal effects of two (or potentially more) risk factors on an outcome and is analogous to factorial RCT); Bi-directional MR (whereby MR, instruments for both exposure and outcome are used to assess the causal association in both directions); Multivariable MR (whereby the researchers aim to estimate the effects of two or more exposures on an outcome); and Two-step MR, (not to be confused with two-sample MR), which allows researchers to use genetic instruments to assess mediation in a potentially causal pathway [40,41].

Standard one-sample MR provides only a single effect estimate, which may not be informative if the effect of the exposure is non-linear. If individual level data are available for a continuous exposure, non-linear MR can be applied to estimate whether the causal effect of the exposure on the outcome varies across different levels of the exposure [42,43]. The phenome-wide association study (PheWAS) approach has been used to investigate the association between a set of genetic variants and a set of phenotypes, testing the association of each genetic variant and phenotype pair individually [44]. The rapidly expanding number of genetic variants with known associations has now facilitated the extension of the PheWAS approach to test for causal relationships using MR [45]. This approach has been called MR-PheWAS and should be viewed as a “hypothesis-generating” or agnostic approach whereby identified potential causal associations need to be followed up and validated using more detailed replication studies [46].

## 6. Selected Recent Applications of Mendelian Randomization in Nutritional Epidemiological Studies

MR studies have become more prevalent in the nutritional literature as the methodology has evolved and statistical resources have become available to facilitate analyses. We now give some exemplars of the types of research question MR has been used to address in nutritional research.

### 6.1. Confirmation or Refutation of an Observational Association

Observational studies of dietary intake and blood concentrations of vitamins E and C, lycopene, and carotenoids have been associated with a lower risk of incident ischemic stroke. Marten and colleagues [47] conducted a two-sample MR to assess the associations between genetically influenced circulating levels of antioxidants and their metabolites with ischemic stroke. SNPs that were genome significant (*p* < 5 × 10^−8^) were selected for vitamins E (11 SNPs), vitamin C (14 SNPs), lycopene (5 SNPs), beta-carotene (3 SNPs) and retinol (24 SNPs). Summary statistics on the associations of the exposure-related SNPs with ischemic stroke were extracted from three large cohorts that included 70,791 cases of ischemic stroke and 987,507 controls of European ancestry. They used IVW regression analysis to ascertain the cohort-specific causal estimate and the results from each cohort were meta-analyzed and several sensitivity analyses were performed to assess the robustness of the findings. It was found that, for absolute antioxidants levels, the odds ratios (ORs) ranged between 0.94 (95% CI, 0.85–1.05%) for vitamin C and 1.04 (95% CI, 0.99–1.08%) for lycopene. There was no evidence of a causal association between dietary-determined antioxidant levels and ischemic stroke. The authors concluded that antioxidant supplements to increase circulating levels are unlikely to be of clinical benefit in preventing ischemic stroke [47]. This is a clear refutation of the evidence provided from observational studies of antioxidants with ischemic stroke.

### 6.2. MR to Overcome Reverse Causality

It is known that high levels of dietary copper intake can lead to deposition of copper in the kidney and to a decline in kidney function [48]. However, the direction of effect is unclear as the relationship between copper and kidney disease is bi-directional as imbalances in the circulating copper levels may also occur as a result of impaired renal excretion or a result of changes in protein metabolism in patients with chronic kidney disease (CKD) [49]. Ahmad et al. [50] conducted a two-sample MR using GWAS estimates for circulating copper measured in both plasma and serum from a meta-analysis of three population-based cohorts comprising 6937 individuals of European-descent. The authors used two genetic variants that were found to be associated with circulating copper levels at genome-wide significant level to estimate the causal relevance of genetically predicted copper with the kidney outcomes. It was found that genetically predicted higher circulating copper levels were associated with higher CKD prevalence (odds ratio 1.17 [95% CI, 1.04–1.32%]) and was weakly associated with decline of the glomerular function rate (odds ratio 1.10; [95% CI, 0.99–1.23%]). The authors concluded that elevated circulating copper levels may be a causal risk factor for CKD and therefore warrants further investigation to clarify the underlying mechanisms and the clinical relevance [50]. However, the authors were unable to rule out potential pleiotropic effects as it was not possible to apply some of the key sensitivity analyses as only two SNPs were available.

### 6.3. MR to Predict Efficacy in the Absence of Trial Evidence

Genetic support can improve the chances that a future RCT will succeed [51]. Colorectal cancer (CRC) is the fourth most common cancer in the UK and worldwide [52]. Although incidence rates among the over 50s have remained relatively stable, rates in younger age groups have increased globally [53]. There is a need to identify potential new treatments as well as complementary prevention strategies to reduce risk of CRC. Salicylic acid (SA) is a dietary metabolite that can be found in various fruits, vegetables, herbs, and spices [54] and there is good observational evidence that it is associated with lower risk of CRC. Aspirin is known to have effects on SA but there are no primary prevention trials to assess the effect of Aspirin or SA intervention on CRC risk. In the absence of RCT evidence, Nounu et al. aimed to use a two-sample MR approach to address whether levels of SA affected CRC risk, stratifying by aspirin use [55]. A two-sample MR analysis was performed using summary statistics of SA (*N* = 14,149) and CRC (55,168 cases and 65,160 controls). There were only between 2–6 SNPs available for pathway analysis (defined as variants within the coding regions of the enzyme involved in aspirin and SA metabolism); and 1–4 SNPs were available from genome-wide significant SNPS and used as instruments for SA [55]. A study of 4410 cases and 3441 controls was used for replication and stratification of aspirin use. The main analyses used IVW and conducted sensitivity analyses for horizontal pleiotropy and the authors found no association between SA and risk of CRC with an OR of 1.08 (95% CI, 0.86–1.34%) [55]. The results remained largely unchanged after stratification for aspirin use. The authors concluded that there was no evidence to suggest that an SD increase in genetically predicted SA protects against CRC risk in the general population and upon stratification by aspirin use, implicating a limited value of taking SA or Aspirin for the purpose of preventing CRC. However there was some evidence of weak instrument bias (F-statistic < 10, indicating that the genetic instruments explained little of the variance in SA levels). The authors acknowledged that their analysis needs to be repeated again with stronger instruments that proxy the metabolite levels in order to draw more robust conclusions [55].

There has been long standing controversy about the hypothesized inverse relationship of role of moderate alcohol consumption with cardiovascular outcomes observed in observational studies. As trials are unlikely to be conducted in this area due to ethical issues, MR studies can play a very important role. Alcohol (ethanol) is a macronutrient metabolized primarily by the liver and the enzyme alcohol dehydrogenase transforms alcohol into the toxic compound acetaldehyde, which gets broken down by aldehyde dehydrogenase into a less toxic compound acetate. There is a functional variant (rs129884) of the alcohol dehydrogenase 1B (ADH1B) gene that is common in Europeans and is associated with higher levels of alcohol consumption. In East Asian populations, a loss-of function variant (rs671) of the aldehyde dehydrogenase 2 (ALDH2) gene, that is very rare in Europeans, leads to decreased acetaldehyde breakdown, high acetaldehyde concentrations, and a flushing reaction following alcohol drinking and thus reduced consumption [56]. Millwood and colleagues conducted both observational and genetic analyses of alcohol consumption using these two variants in an East Asian population. This work found that genetically-predicted higher alcohol levels were positively associated with higher stroke risk and higher blood pressure levels, thus clearly refuting the J-shaped associations often reported from conventional observational analyses [56].

### 6.4. MR for Hypothesis Generation

MR can be used to evaluate or replicate known associations or discover new relationships or generate new hypotheses, for example through use of a ‘phenome-wide scan’, which is facilitated by the availability of large-scale prospective biobanks with long-term follow-up of incident events via linkage to electronic health records, such as the China Kadoorie Biobank (CKB) [57] and UK Biobank [58].

Magnesium is the fourth most abundant mineral in humans [59], and its primary source is food, especially those rich in dietary fiber and dairy products [60]. Both magnesium deficiency and excess magnesium have been associated with several NCDs in epidemiological studies but there is uncertainty about the causal relevance of these associations. Li et al. recently performed MR-PheWAS using the UK Biobank database to investigate the causal relationship of a wide range of disease outcomes with serum magnesium levels [61]. The authors identified 1888 unique International Classification of Diseases coding systems (ICD-10) codes. They found evidence of potential adverse effects of higher magnesium status on risk of female breast cancer, myeloproliferative disease, polycythemia vera, cataract, degenerative skin conditions and other dermatoses, seborrheic keratosis, and fracture of upper limb (e.g., radius and ulna). In contrast they found evidence of a potential beneficial effects of higher magnesium status on risk of gout as well as inguinal hernia [61]. In total, their MR-PheWAS analysis implicated a causal role of magnesium in five disease groups and six disease outcomes. In addition, their study indicated the gender-specific effects of nine disease groups/outcomes in MR estimated effects. As serum magnesium is potentially modifiable, the authors recommended that exploration of approaches to optimize levels of magnesium should be considered for targeted interventions [61].

### 6.5. MR to Potentially Repurpose Nutritional Supplementation Strategies

MR can also be used to inform the potential of repurposing a nutrient supplementation. There is uncertainty about whether low vitamin D status is associated with higher risk of type 2 diabetes or is just a marker of overall poor health. The incidence of type 2 diabetes has increased substantially in both high-income and low- and middle-income countries in recent decades [62]. Vitamin D insufficiency, defined as plasma 25-hydroxyvitamin D (25(OH)D) concentration < 75 nmol/L, is common in European and Asian populations, particularly among those living at high latitude, during the winter months, or among those in cities with poor air quality [63,64]. Observational studies have demonstrated that individuals with higher vitamin D status in European populations have lower risks of developing type 2 diabetes. The Vitamin D Supplementation and Prevention of Type 2 Diabetes trial of 2423 participants for North America (1211 in the vitamin D group and 1212 in the placebo group) found a hazard ratio of 0.88 (95% CI, 0.75–1.04%) after a median follow-up of 2.5 years. A recent randomized trial of active vitamin D treatment for type 2 diabetes in Japanese participants was inconclusive, perhaps due to lack of statistical power [65]. Lu et al. [66] performed a detailed MR study to: (i) examine the associations of genetic scores for 25(OH)D concentration with the 2 synthesis SNPs (DHCR7-rs12785878 and CYP2R1-rs10741657) vs all available SNPs (2 synthesis SNPs plus 1 transport SNP [GC/DBP-rs2282679] and 1 catabolism SNP [CYP24A1-rs6013897]) in 82,464 Chinese adults from the CKB)(57); (ii) to conduct an updated meta-analysis of all genetic studies assessing the effects of genetically instrumented differences in plasma 25(OH)D concentrations on risk of type 2 diabetes in a primary analysis using the two synthesis SNPs and a secondary analysis using all four SNPs for 25(OH)D concentration; (iii) to compare the risks of diabetes associated with equivalent differences in biochemically measured versus genetically instrumented plasma 25(OH)D concentrations. The CKB results were combined in a meta-analysis of 10 studies for the two synthesis SNPs (*N* = 58,312 cases) and seven studies for all four SNPs (*n* = 32,796 cases). Mean (SD) 25(OH)D concentration was 62 (20) nmol/L in CKB, and the per allele effects of genetic scores on 25(OH)D were 2.87 (SE 0.39) for the synthesis SNPs and 3.54 (SE 0.32) for all SNPs. A 25-nmol/l higher biochemically measured 25(OH)D was associated with a 9% (95% CI, 0–18%) lower risk of diabetes in CKB. In a meta-analysis of all studies, a 25-nmol/L higher genetically instrumented 25(OH)D concentration was associated with a 14% (95% CI, 3–23%) lower risk of diabetes (*p* = 0.01) using the two synthesis SNPs [66]. We concluded that trials of vitamin D supplementation may not be able to accrue sufficient numbers of type 2 diabetes cases for reliable assessment of the effect of vitamin D on the risk of type 2 diabetes, and additional trials with larger numbers of such cases may be needed. In addition, we have also suggested that meta-analysis of ongoing and future trials of vitamin D supplementation are required before advocating use of vitamin D supplements for the prevention of diabetes [66]. A recent meta-analysis from another research group found that vitamin D supplementation was associated with a 11% lower risk of diabetes in participants with pre-diabetes [67]. A more recent overview of the health effects of vitamin D supplementation for type 2 diabetes (and other disease outcomes) suggested that additional studies or more in-depth analysis of existing studies are required to validate the findings [68].

### 6.6. MR to Inform the Design of an RCT

It has long been known that *MTHFR* C677T polymorphism is associated with homocysteine levels and also affects how the body process folate. Randomized trials of folic acid to lower homocysteine for the potential prevention of cardiovascular outcomes were largely null [69]. This was attributed to the fact that the RCTs were often conducted in countries where there was voluntary or mandatory fortification of foods with folic acid, primarily for the prevention of neural tube defects. A comprehensive MR study that utilized meta-analysis of the associations of *MTHFR* C677T stroke outcomes confirmed that there was no evidence of benefit for stroke in regions of the world where there was fortification (OR = 0.97, 95% CI, 0.90, 1.11), but there was evidence of potential benefit for stroke in Asia (OR = 0.59, 95% CI, 0.51, 0.69) where there was no folate supplementation [70]. This led to a~20,000 patient trial the (China Stroke Primary Prevention Trial) of folic acid to be designed to target individuals with hypertension but without a prior history of stroke or myocardial infarction in a low-folate population in China. [71] They compared a blood pressure (BP) lowering drug versus the same BP lowering drug plus folic acid and found that the risk of overall stroke was reduced by 21% (95% CI, 7, 32%) in the BP lowering drug plus folic acid group compared to BP lowering alone [71]. However, due to the relatively short duration of the trial, the study was unable to provide definitive evidence on stroke subtypes or on adverse events. Subsequent MR studies of homocysteine have provided strong support for the causal relevance with small vessel disease [72].

## 7. Summary

There has been an excellent recent narrative review of the use of MR for causal inferences in nutritional research [73], but the focus of that review was on the fundamental principles and it did not cover the methodological aspects in detail, so it may not be as useful to a novice wishing to conduct their own MR analyses [73]. There is also a good recent methodological overview of the specific role of MR in nutrition and cancer aimed at cancer researchers [74]. However, this report has endeavored to give a non-technical overview for a nutritional audience of the use of MR methodology to answer different types of research questions for a range of NCDs. There are also many good non-technical overviews that target clinicians [75,76,77] and some more technical overviews for a more general health science audience [37,78,79]. There is now a large body of literature on how to read [80], conduct [81,82] and report [83] MR studies. Well-conducted, reported and reliable MR studies can be extremely valuable and cost-effective for nutritional research, but more work needs to be done in order to identify reliable genetic instruments for nutritional factors, such as nutrient intake. Importantly, due to the nature of nutritional factors (e.g., macronutrients are often highly interrelated and may affect multiple pathways), there is likely to be horizontal pleiotropy and confounding that can make MR either unfeasible or more challenging than in other areas of epidemiology. It may be necessary to employ more sophisticated MR designs such as multivariable MR or two-step MR to overcome some of these issues [84]. Weak instrument bias in MR studies of nutritional factors can also make it more difficult to ascertain robust findings and are likely to require international large-scale collaborative efforts in order achieve sufficient statistical power. In addition, in common with other specialty areas, there need to be more MR studies in non-European populations in order to obtain a more diverse evidence-base. Despite, these caveats, the availability of publicly available summary statistics from large-scale genetic consortia provide opportunities to assess the causal relevance of nutritional exposures, which may lead to potentially more successful intervention studies for non-communicable diseases.

## Figures and Tables

**Figure 1 nutrients-14-03408-f001:**
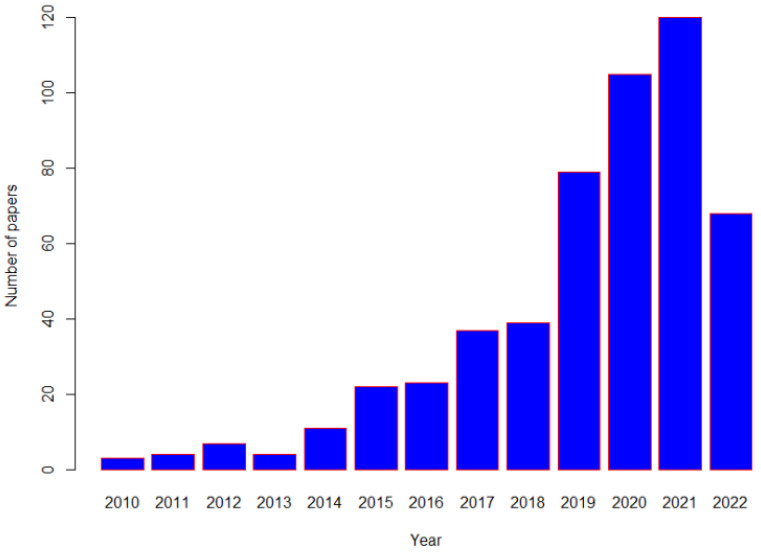
Number of published Mendelian Randomization studies related to nutrition in Pubmed from 2010 up to 10 June 2022.

**Figure 2 nutrients-14-03408-f002:**
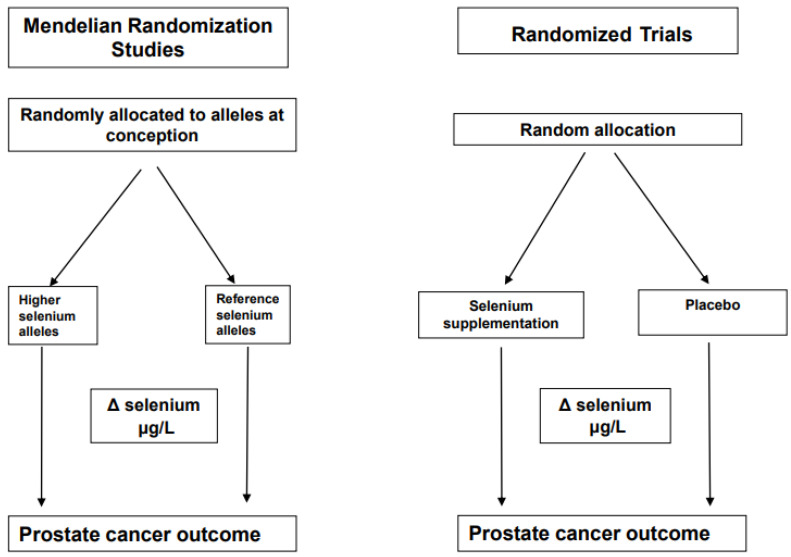
Comparison of a conventional trial with a Mendelian Randomization study. This illustrates the analogy between a conventional randomized controlled trial and a Mendelian randomization study. Δ represents the change.

**Figure 3 nutrients-14-03408-f003:**
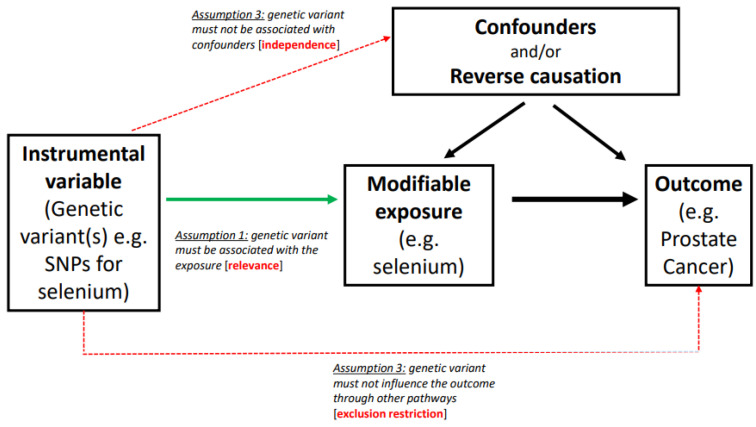
Illustration of the three key assumptions of Mendelian Randomization studies. This illustrates the relevance, independence and exclusion restriction assumptions of Mendelian Randomization with selenium as the modifiable nutritional exposure. The relevance assumption can be easily tested, and is considered as fulfilled if the SNP-exposure association has an F-statistic > 10. The independence assumption is hard to validate as problems due to pleiotropy and population substructure may occur but associations with known confounders should be null. In general, the exclusion restriction assumption is hard to validate as there may be pleiotropic effects of SNPs or SNPs in linkage disequilibrium correlated with genes that have effects on the outcome independently of the exposure. It is important to perform a variety of sensitivity analyses that make different assumptions about pleiotropy.

**Figure 4 nutrients-14-03408-f004:**
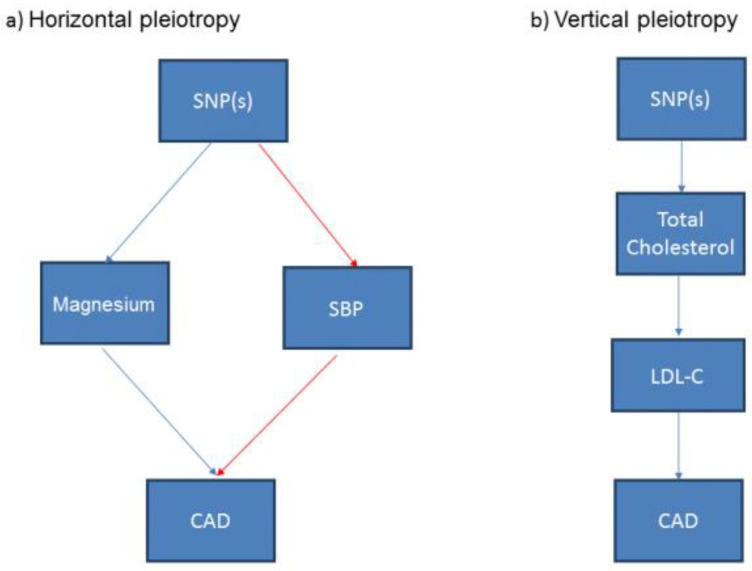
Simplified illustration of (**a**) horizontal and (**b**) vertical pleiotropy in Mendelian Randomization in nutritional research. This illustrates that (**a**) horizontal pleiotropy occurs when the SNPs have effects on multiple exposures that are independent of each other; (**b**) vertical pleiotropy occurs where the effects of one exposure can have a downstream impact on another related exposure. SNP: Single nucleotide polymorphism; SBP: Systolic blood pressure; LDL-C: Low density lipoprotein cholesterol; CAD: Coronary Artery Disease.

**Table 1 nutrients-14-03408-t001:** Some examples of nutritional biomarkers.

Proposed Biomarker	Type of Biological Sample	Nutritional Assessment
Carotenoids	Plasma	Fruit and vegetable intake
Creatine	Serum	Meat and fish intake
Dyhydrocaeic acid	Urine	Coffee intake
Homocysteine	Plasma	Folate status
Pentadecanoic acid	Plasma/serum	Total dairy fat intake
25-hydroxyvitamin D	Plasma/serum	Vitamin D intake
Caffeine	Plasma	Caffeine intake

**Table 2 nutrients-14-03408-t002:** Some selected useful statistical software resources for Mendelian Randomization.

Name of Resource	Notes	Weblink
One-sampleMR	R package for one-sample MR	https://remlapmot.github.io/OneSampleMR/ (accessed on 12 July 2022)
ivmodel	R package that fits instrumental variable analyses for individual data	https://cran.r-project.org/web/packages/ivmodel/ivmodel.pdf (accessed on 12 July 2022)
ivonesamplemr	Stata function for implementation of one-sample MR	https://github.com/remlapmot/ivonesamplemr (accessed on 12 July 2022)
glsmr	R package that can be used to perform a non-linear (stratified) one-sample MR analysis	https://rdrr.io/github/hughesevoanth/glsmr/man/glsmr.html (accessed on 12 July 2022)
Two-sampleMR	R package for two-sample MR analysis, directly links to MR-Base database	https://github.com/MRCIEU/TwoSampleMR/ (accessed on 12 July 2022)
MendelianRandomisation	R package for two-sample MR analysis, links to Phenoscanner * database	https://cran.r-project.org/web/packages/MendelianRandomization (accessed on 12 July 2022)
MR Robust	Stata package for two-sample MR analysis	https://github.com/remlapmot/mrrobust/ (accessed on 12 July 2022)
MR-Base	GWAS summary database of more than 1100 GWAS studies and online platform to automate two-sample MR	http://www.mrbase.org/ (accessed on 12 July 2022)
MR-SENSEMAKR	A suite of sensitivity analysis tools that quantify both how much the inferences would have changed under a postulated degree of violation, as well as the minimal strength of violation necessary to overturn a certain conclusion of an MR	https://doi.org/10.5281/zenodo.5635471 (accessed on 12 July 2022)
PHEASANT	R package for performing phenome scans in UK Biobank, including MR phenome-wide association studies (MR-pheWAS)	https://github.com/MRCIEU/PHEASANT/ (accessed on 12 July 2022)

* PhenoScanner is a curated database holding publicly available results from large-scale genome-wide association studies. (http://www.phenoscanner.medschl.cam.ac.uk/about/, accessed on 12 July 2022).

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
