# Peer review of "An Overview of Methods and Exemplars of the Use of Mendelian Randomisation in Nutritional Research"

_nutrients, 2022, doi:10.3390/nu14163408_

Round 1
Reviewer 1 Report
Excellent review of a booming topic. Sometimes it is difficult to understand given the specificity and depth of the subject. Although for the readers it can be a starting point for the knowledge of this area.
Author Response
We thank the reviewer for their comments. We have tried to limit the technical content of the manuscript and have referred the interested reader to reports that consider some of the more nuanced and technical issues whenever possible. reports
Reviewer 2 Report
This short review paper is overall well written but a bit too superficial. I would suggest to expand the examples somewhat.
Specific comments:
1) Table 1. Should you also add (or replace) plasma caffeine as a marker of coffee consumption? You mention caffeine as a biomarker for coffee in the text (page 4).
2) I would suggest to remove Table 2 and place the information from this table in the text and/or as a footnote to Figure 3.
3) The examples provided are very few. I would suggest to add some more examples, for example mention MR results for folate (and possibly other B vitamins) and selected outcome(s), such as stroke and cancer, and compare it with findings from observational and RCTs. In this case, findings for folate and small vessel stroke are interesting as results from MR studies and RCTs are broadly consistent.
4) What about MR studies on other important vitamins, such as vitamin A (retinol) and vitamin E (alpha-tocopherol), and major minerals, such as calcium, potassium, and sodium? Could you give some examples. For vitamins A and E and possibly other nutritional exposures it would be good to mention that the genetic variants are few.
5) I would also suggest to mention macronutrients other than alcohol as well as dietary patterns and specific foods. The associations of those nutritional exposures and diseases are difficult to assess using the MR design.
6) In the examples provided, please mention how many SNPs that were used as instrumental variables for the nutritional exposure in the MR analysis.
Round 2
Reviewer 2 Report
The manuscript has overall been satisfactory revised.
Minor edits are needed:
1) "causal" is misspelled to "casual" at several places in the text
2) In section "6.6. MR to inform the design of and RCT", the references have not come through correctly